# Seasonal variations in household food security and consumption affect women's nutritional status in rural South Ethiopia

**Bethelhem Mezgebe** *, **Taye Gari, Mehretu Belayneh , Bernt Lindtjørn**

School of Public Health, College of Medicine and Health Sciences, Hawassa University, Hawassa, Ethiopia

* bethelhem_mezgebe@yahoo.com

## Abstract

Food availability varies seasonally in most rural areas of developing nations, especially in areas affected by drought and climate change, with women being one of the most vulnerable groups. This study aimed to assess the effect of seasonal variation in household food security, adequate dietary diversity, food consumption, and wealth on the nutritional status of women of reproductive age in a rural community in South Ethiopia. Further, the study aimed at identifying associated factors with women's nutrition status. An open cohort study was conducted from June 2021 to June 2022, with follow-up visits every three months. Anthropometric measurements were carried out along with interviews. Data were analyzed using STATA version 15. Multilevel, multiple linear regressions were employed. Findings revealed that women's average body mass index (BMI) was 20.4 kg/m$^2$ (95% CI: 20.4–20.5). The highest (20.6 kg/m$^2$, 95% CI: 20.5–20.8) was observed in December, while the lowest (20.2 kg/m$^2$, 95% CI: 20–20.3) occurred in September. During the main postharvest period in December, the household food insecurity score was the lowest (median: 4, Inter quartile range (IQR): 0–9), while the household dietary diversity score (median: 6, IQR: 5–7), and the household food consumption score were the highest (median: 50.5, IQR: 44–70). Factors such as household food security, food consumption, previous season BMI, age, marital status, and membership in safety net programs were identified as determinants of women's BMI. The study showed the vulnerability of women in drought-prone areas to seasonal undernutrition. We recommend collaborative work among stakeholders to ensure sustainable food access and minimize seasonal food shortages' effect on women's nutrition and overall well-being.

## Introduction

Farmers in low-income countries often rely on rain-fed agriculture, which is affected by seasonal and weather variability [1, 2]. This can lead to low and insufficient agricultural production for year-round consumption, resulting in periodic food scarcity, as shown in countries such as Burkina Faso, Ghana, Bangladesh, and Ethiopia [3–6]. Seasonal food shortages can also result in unstable household wealth and changing household food security [7–11], often affecting women's nutritional status [12–14].

**Data Availability Statement:** The data is deposited on Dryad with dataset title "Household food accessibility and women nutrition in rural South

Ethiopia". DOI https://doi.org/10.5061/dryad.sj3tx96c1.

**Funding:** This study was funded by Norges Forskningsråd or the Research Council of Norway (CMI Project: 18033) in the form of funding awarded to the Co-producing Gender-responsive Climate Services for Enhanced Food and Nutrition Security and Health in Ethiopia and Tanzania (COGENT) project conducted under Hawassa University. The funders were not involved in the study design, data collection and analysis, publication decision, or manuscript preparation.

**Competing interests:** The authors have declared that no competing interests exist.

The effect of seasonal food shortages on women's nutrition can be influenced by other factors such as workload, illness, culture, and reproductive responsibilities. Women usually work from dawn to dusk throughout the year on household chores, social obligations, paid labor, and substantial agricultural tasks during peak seasons [15, 16]. This heavy workload and inadequate food during lean periods can lead to an energy imbalance and undernutrition [17–20].

Illness is the other contributing factor to undernutrition, creating a vicious cycle [21]. Illness results in a loss of appetite, diminished nutrient absorption, and increased energy expenditure, all collectively contributing to undernutrition [22]. Undernutrition, in turn, can increase the incidence and severity of illnesses [23]. This can negatively influence household wealth by limiting women's productivity and increasing healthcare expenditure [24]. Studies in Ethiopia showed that intestinal parasitic infestation [25], anemia [26], malaria, and other infectious diseases can predict women's undernutrition. Limited decision-making power, a lack of money for treatment, and a lack of access to quality healthcare exacerbate the impact of illness on women's nutritional status [27, 28].

Culture can impact women's undernutrition by influencing the amount and quality of food they eat [29]. Pregnancy and breastfeeding increase nutritional needs, but studies show women in several countries often don't meet these recommendations, especially during periods of food shortages [8, 13, 30–32]. Focusing on women's nutrition is essential because they are vital to household food security since they are frequently involved in food production, purchasing, preparation, preservation, and allocation [33]. Furthermore, women often invest their income to improve the health and education of their children [19]. Therefore, improving women's nutrition is about her and the whole family's well-being.

Previous research in Nepal, Madagascar, and Ethiopia demonstrated the existence of seasonal variation in household food security, dietary diversity, and food consumption [13, 34–36]. However, the effect of these variations on the nutritional status of women of reproductive age group has yet to be adequately investigated. Many nutrition-related studies in Ethiopia and other low-income countries usually focus on pregnant women or mothers with children under the age of five, excluding women who are not yet mothers or are unable to conceive [37–40]. It is essential to recognize that assessing and improving women's nutrition extends beyond reproductive responsibilities. In Africa, women make up nearly 40% of the agricultural labor force [41], highlighting their significant contribution to the economy and emphasizing the importance of their well-being.

In our study, we conducted a more comprehensive assessment of the wealth index compared to the usual practice of measuring it only once. Instead, we measured it three times within a year to capture its temporal variation. Additionally, we considered the impact of women's previous BMI on their current BMI.

This study aimed to assess the effect of seasonal variation in household food security, adequate dietary diversity, food consumption, and wealth status on the nutritional status of women of reproductive age in a rural community in South Ethiopia and identify associated factors with women's nutritional status. To achieve these objectives, we tested the hypothesis that women's body mass index remains unchanged regardless of seasonal variations in household food security, consumption, dietary diversity, and wealth.

## Methods and materials

### Study area description, food production, and food availability

An open cohort study was conducted in the Boricha and Bilate Zuria districts in the Sidama National Regional State from June 2021 to June 2022, with follow-up visits every three months.

The Sidama Region has 30 districts and 636 *kebeles* (the smallest administrative unit in Ethiopia). Bilate Zuria and Boricha are drought-prone districts with 30 rural and two urban

*kebeles*. In 2022, the population size in these two districts was estimated to be 282,002, with 142,214 females. The annual birth rate was 22.8/1000 population, and women of reproductive age constitute 23.3% of the population. One primary hospital, eight health centers, and 30 health posts are governmental health institutions that provide health services to the population in the two districts [42].

The districts have a bimodal rainfall pattern; the main is *Belg*, which lasts from March to May, and the other rainy season, *Kiremt*, lasts from June to the end of September. Despite the presence of Bilate River in Bilate Zuria, both districts experience water scarcity, particularly during dry seasons [43].

The area has two harvesting times; the first harvest usually begins in August for short-cycle crops like potatoes, sweet potatoes, pepper, and haricot beans. The second harvest time starts around October for long-cycle crops like maize [44]. Enset (*Ensete ventricosum*), which takes 5 to 7 years to mature, is a resilient food source during lean periods [45]. Kale and sugarcane usually grow throughout the year. Some residences cultivate perennial crops like coffee, khat (*Catha edulis*), banana, and avocado. Coffee, khat, sugarcane, and paper are cash crops in the area.

Boricha and Bilate Zuria districts are food-insecure areas. The food shortage gradually improves with the first harvest as household food stocks replenish and mini corn (unripe corn) bridges the food deficit until the second harvest. Food stocks reach their peak during the second harvest [46]. However, in March, the food stocks usually deplete, and household food availability declines.

Food insecurity is deeply rooted and has multiple underlying causes in the area. Despite efforts to improve livelihoods, household food insecurity remains a persistent problem [47, 48]. Population density, land scarcity, limited crop diversity, traditional farming methods, and climate change contribute to decreased crop and livestock production, exacerbating household food insecurity [49–51].

Furthermore, the COVID-19 pandemic, national instability due to the civil war and local disputes have added to the challenges faced by the residents. These conditions have led to price inflation for fertilizers, seeds, and food, worsening the already vulnerable situation.

## Ethical consideration

The Institutional Review Board of Hawassa University provided ethical clearance for the current study, conducted under the "Co-producing Gender-responsive Climate Services for Enhanced Food and Nutrition Security and Health in Ethiopia and Tanzania (COGENT) project." The project team obtained the first clearance on May 20, 2021(reference number IRB/ 181/13). Subsequently, a second clearance was obtained specifically for assessing women's nutritional status, illness assessment, and level of physical activity on September 7, 2021 (reference number IRB/275/13). Additionally, the project team acquired the necessary permission letters from the Regional Health Bureau and district health offices.

Written informed consent was obtained directly from participants who were 18 years old and above. In the case of adolescents below the age of 18, written consent was obtained from their parents or legal guardians. Throughout the study, strict confidentiality measures were implemented, ensuring that all information remained confidential. Personal identifiers were not used during the data analysis and data-sharing processes.

## Data collection procedure

The study included households that had previously participated in a study conducted in 2017 [34]. To select the participants, a multistage sampling technique was employed. From the 30

rural *kebeles* in the districts, nine *kebeles* were randomly chosen. Subsequently, a total of 910 households were selected using cluster sampling. All eligible girls and women between the ages of 15 and 49 from the selected households were included in the study, except those who were seriously ill, paralyzed, deformed or belonged to families experiencing grief or festivity during the data collection period.

The sample size was calculated using OpenEpi version 3.03 (Open Source Epidemiologic Statistics for Public Health; www.OpenEpi.com). The sample size for the study was calculated based on the seasonal variation of women's BMI. A sample size of 904 women was determined using assumptions from a previous cohort study conducted in the eastern part of Ethiopia [14]. In that study, the percentage of mothers with a BMI less than 18.5 kg/m$^2$ during the pre-harvest season was 54.7%, and the percentage during the post-harvest period was 41.7%. A confidence level of 95% and a power of 80% were considered, with a ratio of unexposed to exposed individuals set at 1:1. Considering a design effect of 1.5 and assuming a 20% non-response rate; the final sample size was estimated to be 904 women. However, to minimize the effect of mobility and non-respondent rate we include all eligible (1,089 women) in the selected households.

## Outcome variables

The outcome variable was body mass index (BMI). Body weight was measured using a digital *SECA* scale (SECA GmbH, Germany) with a precision of 0.1 kg. The weight measurements were conducted using the following procedure: Before data collection, we verified the accuracy of the weight scales using an object with a known weight. Data collectors ensured the scale reading was exactly at zero ahead of any measurement, and the measurements were taken on a flat surface. The individuals being weighed wore lightweight clothes and stood barefoot.

Height was measured with a standard stadiometer (SECA 217, GmbH, and Study Germany) to the nearest 0.1 cm. The participants stood upright, facing forward, with their heads and feet bare. They positioned themselves with their arms hanging at their sides, feet together, and ensuring contact between their heels, buttocks, shoulder blades, and head with the vertical surface of the stadiometer.

*Predictor variables*. Household level predictors were the socio-demographic status of the household head (marital status, educational status, and occupation), family size, household food security, household dietary diversity, household food consumption, household wealth index, and membership in a productive safety net program. Individual level factors were socio-demographic characteristics of the women (age, marital status, educational status, and occupation) and previous season BMI.

*Food consumption score*. Household food consumption was evaluated using a food frequency questionnaire comprising 12 food groups [52]. The questionnaire assessed the frequency at which households consumed specific food groups. Respondents were provided with eight options to recall the frequency of consumption. 1: once/week, 2: twice a week, 3: 3 times a week, 4: 4 times a week 5: 2 or more times a month 6: once a month 7: yearly 8: never

To calculate the food consumption score, the weekly consumption of each food group was weighted based on its nutrient density. The weighting scheme was as follows: cereals and roots were combined as main staples and assigned a weight of 2, vegetables weighed 1, fruits had a weight of 1, meat, fish, eggs, and milk had a weight of 4, pulses weighted 3, sweets weighted 0.5, fats weighted 0.5, and condiments weighted 0 [52]. The weight assigned to each food group was multiplied by the number of days that the food group was consumed in a week. These weighted scores for each food group were then summed together to obtain the total food consumption score. This score was classified based on the following cut points [52]:

Households with a food consumption score $< = 21$: poor food consumption;

Households with a food consumption score of 21.5 to 35: borderline food consumption; and,

Households with a food consumption score $> 35$: acceptable food consumption.

*Household dietary diversity*. A household dietary diversity score was calculated by summing up the number of food groups eaten in the house during the day and night before the date of data collection based on 24 hours of dietary recall for 12 food groups that include cereals, roots, or tubers; vegetables; fruits; meat or poultry; eggs; fish and seafood; pulses or nuts; milk and milk products; oil or fats; sugar or honey; and condiments or coffee [53]. Households with a household dietary diversity score $< = 3$ were considered to have low household dietary diversity, 4 to 6 medium household dietary diversity, and $> = 7$ high household dietary diversity.

*Household Food Insecurity Access Scale (HFIAS)*. The Household Food Insecurity Access Scale (HFIAS), validated for use in the Ethiopian setting, was used to evaluate the seasonal household food insecurity variability [54–56]. It determined the household food insecurity condition four weeks before the study. It contains nine conditions, and each of them was scaled as never = 0, rarely = 1 (1 or 2 times a month), sometimes = 2 (3 to 10 times in a month), and often = 3 (more than ten times in a month). The household food insecurity score was calculated by adding the frequency for each of the nine conditions (0 to 3). The numeric outcome ranges from 0 to 27. The household food insecurity level was classified into food secure, mildly, moderately, and severely food insecure based on the indicator guide mentioned above [54–56].

*Meal frequency*. To assess meal frequency, we utilized a 24-hour dietary recall questionnaire. During data collection, participants were asked yes/no questions regarding the types of meals they had consumed on the previous day. Based on their responses, meal frequency was valued as 0: if nothing had been eaten, 1 for one meal/day, 2 for two meals/day, and 3 for three meals or more /day. They were also asked about the type of meal they had eaten in the past 24hours.

*Wealth index*. Economic status was assessed three times: in June 2021, December 2021, and June 2022. The wealth index was constructed using principal component analysis. Initially, the factors analysis included the following 15 asset-assessing variables: the size of farming land, livestock ownership, electricity, mobile telephone, chair, bed with cotton matters, kerosene-based lamp, type of floor, type of roof, type of wall, place of cooking, telephone expenditure per month, monthly i*dir* (traditional cooperation and the supporting system usually during grief) expense, monthly saving, and participation in *equib* (traditional saving association). The principal component analysis excluded variables with sample adequacy $< 0.5$, communality $< 0.5$, and complex structure. The correlation matrix for the variables in the PCAs contains two or more correlations of 0.30 or greater. The final component analysis included the following five variables: mobile phone, type of roof, type of floor, type of wall, and telephone expense per month. The overall sampling adequacy Kaiser-Meyer-Olkin (KMO) test was $> 0.50$ (0.65), and the P-value of Bartlett's Test of Sphericity was $< 0.001$ [57].

The data collection tool was prepared in English and translated into *Sidaamo Afoo*. It contains questions regarding socio-demographic characteristics, housing conditions, other asset-assessing questions, household food insecurity, 24-hour dietary recall, and food frequency. Data was obtained through face-to-face interviews and anthropometric measurements. Eighteen data collectors, two from each *kebele*, were involved in data collection. All of them had completed tenth grade or more and were fluent speakers of the local *Sidaamo Afoo* language.

Pre-tests were done in the Qorangoge *kebele* found out of the selected *kebeles*. Training was provided for data collectors and supervisors on the questionnaire's contents and how to take anthropometric measurements. To improve the data accuracy, the height of the women was

taken in two seasons, in September 2021 and December 2021, and the average height was taken for BMI calculation. If there were discrepancies between the height measurements, a third measurement was taken. All data collection was closely monitored and supervised.

Data were collected four times in addition to the baseline data: in September and December (post-harvest time) and in March and June (pre-harvest time). Socio-demographic status was assessed only once on baseline (June 2021); household asset (3,133 observations) was collected three times (June 2021, December 2021, and June 2022); household food security (5,238 observations) and household dietary diversity (5,225 observations) were assessed five times throughout the data collection time; food consumption (4,185 observations) and women's anthropometry (3,911 observations) were measured four times: September 2021, December 2021, March 2022, and June 2022.

At the baseline, women from 910 households were recruited. In the final analysis, 1,089 women from 894 households were included. It was observed that 169 households had two or more women participating in the study. At the beginning of the study, there were 1,055 women, and an additional 34 participants were enrolled in September and December. Consequently, a total of 1,089 women participated in the study. On average, 1,050 women of reproductive age group were followed up for 12 months with a follow-up visit every three months (Fig 1).

## Data processing and analysis

The questionnaires were checked and edited during data collection at the end of each day. The data was entered into EpiData version 3.1 (EpiData Classic, Http://www.epidata.dk) and analysed using SPSS Version 25 (IBM Corp, Armonk, NY) and STATA Version 15 (Stata Corp, Texas, USA).

Descriptive statistics was performed to explore the data, identify outliers, and understand the data's nature. Multilevel mixed-effect linear regression was conducted to understand the *kebele*, *Gout* (a subdivision of *kebele*), and household-level variability [58]. In our analysis, the clustering effect at the empty model's household, *Gout*, and *kebele* levels was 65.4%, 3.6%, and 2%, respectively. However, our final model showed no household-level effect, so the analysis was performed without controlling these potential clustering effects. Despite the absence of clustering, we employed multilevel mixed-effect linear regression to address the issue of dependency and unbalanced measurements.

Multicollinearity was checked using variance inflation factor (VIF), normality was assessed through histograms and normality curves, and linearity was examined using a Q-Q plot. Both household and individual level variables with a P-value of $< 0.25$ in the univariate analysis were included in the multivariate analysis. Subsequently, a model was constructed, and the model with the lowest Akaike's information criterion (AIC) value of 5,217 was chosen as the final model. In the multivariable analysis, variables with a P-value of $< 0.05$ were considered the final predictor of women's BMI.

## Results

### Baseline socio-demographic characteristics of participants

Adolescents 15 to 19 years old contribute to 135/1089 (12.4%) of the participants.

The average family size (SD) was 5.6 (1.7) persons. The mean age of the women (SD) was 28.9 (7.6) years. Nearly 80% of the respondents (865/1089) were married. Most women, 71.5% (779/1,089), were engaged in farming as their occupation. Among the participants, 57.7% (628/1089) had not attended formal education. Almost all residents, 99.5% (1083/1089), belonged to the Sidama ethnicity. Additionally, 85.1% (927 out of 1089) of the participants were Protestant.

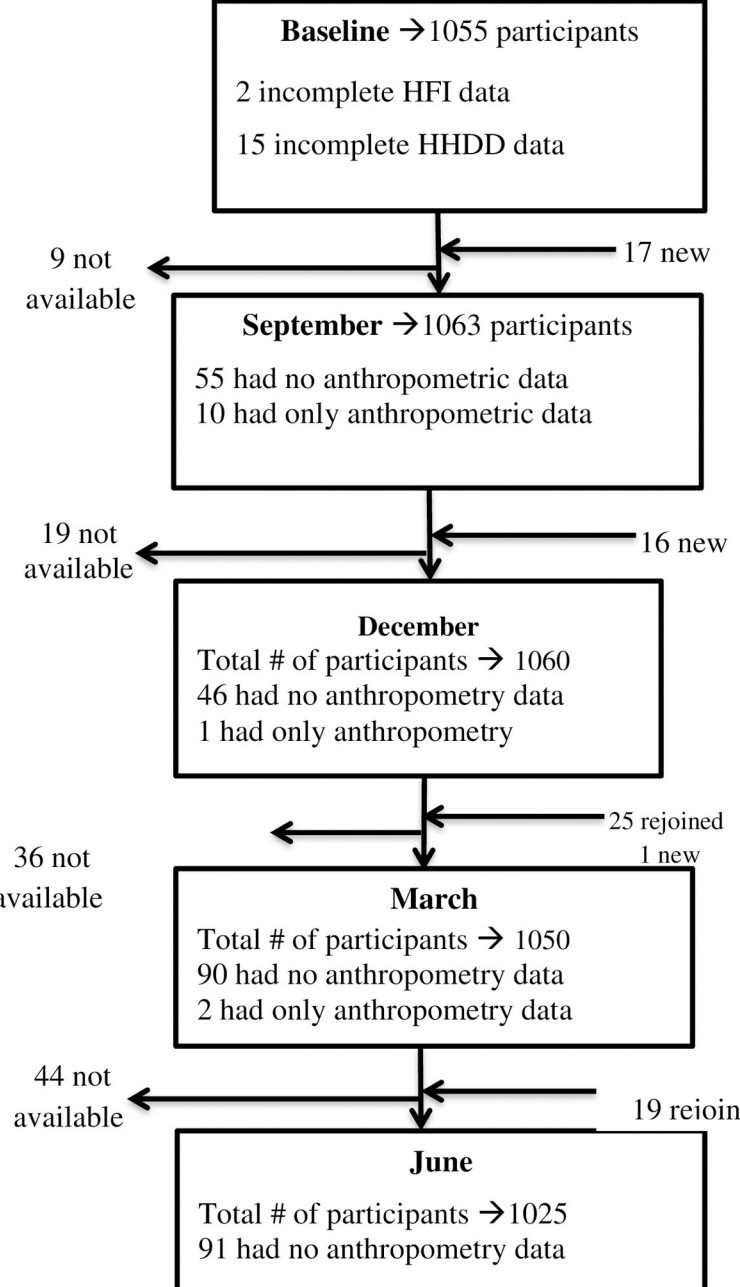

**Fig 1. Flow chart showing the dynamics of study participants.** Flowchart of Study Participants Dynamics. This figure illustrates the flow of study participants across the different phases of the research. It conveys information about the number of participants at each stage, as well as the dynamic movement of individuals in and out of the study. The figure also showed the missing data of main variables in each season. The abbreviations used are: HFI: Household Food Insecurity, HHDD: Household Dietary Diversity.

## Household food security, dietary diversity, meal type, and food consumption

Of 5,238 observations, 15.4% (808 observations) were from food-secured households, while 61% (3,194 observations) were from severely food-insecure households. Regarding the proportion of women from food-secured households, the highest proportion was observed in

**Table 1. Seasonal variation of household food security, dietary diversity, food consumption and meal type among women of reproductive age groups in South Ethiopia: June 2021 to June 2022.**

| Household food security category | June 2021 N = 1053 n (%) | September N = 1053 n (%) | December N = 1059 n (%) | March N = 1048 n (%) | June 2022 N = 1025 n (%) | Over all observation N = 5238 |
|---|---|---|---|---|---|---|
| Food secure | 133 (12.6) | 131 (12) | 347 (32.8) | 130 (12.4) | 67 (6.5) | 808 (15.4) |
| Mildly insecure | 42 (4.0) | 67 (6.4) | 195(18.4) | 139 (13.4) | 82 (8) | 525 (10.2) |
| Moderately insecure | 70 (6.7) | 141 (13.6) | 185 (17.5) | 147 (14) | 168 (16.4) | 711 (13.6) |
| Severely insecure | 808 (76.7) | 714 (67.9) | 332 (31.4) | 632 (60.3) | 708 (69.1) | 3194 (61) |
| **Household Dietary diversity** | **N = 1040** | **N = 1053** | **N = 1059** | **N = 1048** | **N = 1025** | **N = 5225** |
| Low | 123 (11.8) | 217 (20.6) | 54 (5.1) | 95 (9.1) | 57 (5.6) | 546 (10.5) |
| Medium | 704 (67.7) | 725 (68.9) | 721 (68.1) | 779 (74.3) | 854 (83.3) | 3783 (72.4) |
| High | 213 (20.5) | 111 (10.5) | 284 (26.8) | 174 (16.6) | 114 (11.1) | 896 (17.2) |
| **Meal type per day** | **N = 1040** | **N = 1053** | **N = 1059** | **N = 1048** | **N = 1025** | **N = 5225** |
| > = 3 meals | 526 (50.6) | 564 (53.6) | 851 (80.4) | 711 (67.8) | 634 (61.9) | 3286 (62.9) |
| Breakfast and Dinner | 382 (36.7) | 373 (35.4) | 185 (17.5) | 262 (25) | 357 (34.8) | 1559 (29.8) |
| Lunch and Dinner | 100 (9.6) | 59 (5.6) | 17 (1.6) | 63 (6.0) | 29 (2.8) | 268 (5.1) |
| Breakfast and lunch | 6 (0.6) | 57 (5.4) | 5 (0.5) | 12 (1.1) | 2 (0.2) | 82 (1.6) |
| < 2meals | 26 (2.4) | 0 (0.0) | 1 (0.1) | 0 (0.0) | 3 (0.3) | 30 (0.6) |
| **Food consumption** | | **N = 1053** | **N = 1059** | **N = 1048** | **N = 1025** | **N = 4185** |
| Poor | | 81 (7.7) | 27 (2.5) | 28 (2.7) | 15 (1.5) | 151 (3.6) |
| Border line | | 332 (31.5) | 89 (8.4) | 159 (15.2) | 363 (35.4) | 943 (22.5) |
| Acceptable | | 640 (60.8) | 943 (89.1) | 861 (82.1) | 647 (63.1) | 3091 (73.9) |

December, accounting for 32.8% (347 out of 1,059 observations), while the lowest was observed in June 2022, with only 6.5% (67 out of 1,025 observations); see Table 1.

Most households had a medium dietary diversity of 72.4% (3,783/5,225 measurements). In December, the highest percentage of households with a high dietary diversity score was 26.8% (284/1059). Conversely, the lowest, 10.5% (111/1059), was observed in September; see Table 1.

About one-third of the study participants, 37.1% (1939/5225 measurements), reported having two or fewer meals daily. At the baseline, around half of the women (49.4%; 514/1040) skipped one or more meals daily, the highest of all observed seasons. Lunch was the most frequently missed meal (29.8%; 1559/5225); see Table 1.

From all observations, 151 of 4,185 measurements (3.6%) had poor household food consumption. The percentage of households with acceptable food consumption was higher in December at 89.1% (943/1,059) and March at 82.1% (861/1048), compared with September at 60.8% (640/1,053) and June at 63.1% (647/1025); see Table 1.

The household food insecurity score (HFIASS) was the lowest in December (median: 4, IQR (interquartile range): 0–9) and the highest in June (median: 11, IQR: 7–16). The household dietary diversity score was nearly similar during the study periods, with a slight increment in December (median: 6, IQR: 5–7). On the other hand, the food consumption score was highest in December (median: 50.5, IQR: 44–70) and lowest in September (median: 39.5, IQR: 26–53) (Fig 2).

## Wealth index

The median (IQR) wealth index for all observations was 2,211 (900.5, 2,211). From all observations, nearly half of the households (52.1%: 1,635/3,138) were poor. The proportion of women from extremely poor households was the highest in June 2,022 (18.8%: 193/1027). At baseline, the proportion of very poor households (26.2%: 276/1,053) was the highest, while the proportion of women from rich households (11.4%: 120/1053) was the lowest, see Table 2.

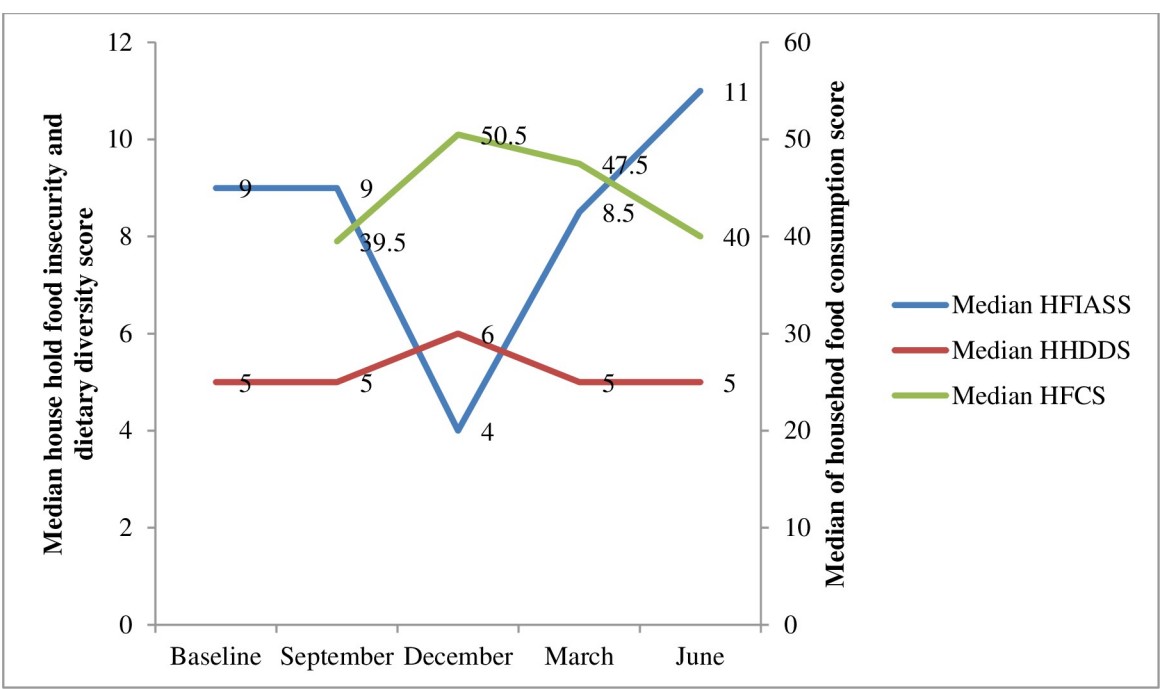

**Fig 2. Seasonal variation in the median score of household food insecurity, dietary diversity, and food consumption of women.**
Seasonal variation in Household Food Access. This figure illustrates the seasonal variations in key measures of household food access. Specifically, it plots the median values and interquartile ranges for: • HFIASS: Household Food Insecurity Access Scale Score.• HDDS: Household Dietary Diversity Score.• HFCS: Household Food Consumption Score.

## BMI of women

The mean BMI (95% CI) for women was 20.4 kg/m$^2$ (20.4, 20.5), with the highest BMI observed in the age group of 20–24 years at 20.9 kg/m$^2$ (20.7, 21.1). Among the different marital status groups, single women had the highest mean BMI of 20.7 kg/m$^2$ (20.5, 20.9). On the other hand, women with no formal education had the lowest mean BMI of 20.2 kg/m$^2$ (20.1, 20.3); See Table 3.

The average BMI of December 20.6 kg/m$^2$ (95% CI: 20.5, 20.8) and March 20.6 kg/m$^2$ (95% CI: 20.4, 20.7) was significantly higher than the average BMI of September 20.2 kg/m$^2$ (95% CI: 20.0, 20.3) and June 20.3 kg/m$^2$ (95% CI: 20.2, 20.5); See Fig 3.

## Predictors of women's nutritional status

A multilevel linear regression was employed to identify factors associated with women's BMI. In the bivariate analysis, the following household-level factors showed a P-value of < 0.25: household food security, dietary diversity, food consumption, and membership in a productive safety net program. Even if the wealth index had a higher P-value, we included it in the final

**Table 2. Household wealth status of women of reproductive age group in June 2021, December 2021, and June 2022, South Ethiopia.**

| Household wealth | June 2021 N = 1053 n (%) | December N = 1058 n (%) | June 2022 N = 1027 n (%) | Overall N = 3138 n (%) |
|---|---|---|---|---|
| Extremely poor | 138 (13.1) | 146 (13.5) | 193 (18.8) | 477 (15.2) |
| Very poor | 276 (26.2) | 184 (17.4) | 148 (14.4) | 608 (19.4) |
| Poor | 519 (49.3) | 579 (54.7) | 537 (52.3) | 1635 (52.1) |
| Rich | 120 (11.3) | 149 (14.5) | 149 (14.5) | 418 (13.3) |

**Table 3. Mean BMI of women 15 to 49 years old in south Ethiopia: September 2021 to June 2022.**

| Variable | Mean BMI (95% CI) | SD |
|---|---|---|
| **Age in year** | | |
| 15–19 | 20.4 (20.2, 20.6) | 2.0 |
| 20–24 | 20.9 (20.7, 21.1) | 2.2 |
| 25–29 | 20.5 (20.4, 20.6) | 2.1 |
| 30–34 | 20.3 (20.2, 20.5) | 2.3 |
| 35–39 | 20.4 (20.2, 20.5) | 2.3 |
| > = 40 | 20.2 (19.9, 20.4) | 2.5 |
| **Relation Ship** | | |
| Household head | 20.3 (19.9, 20.8) | 2.3 |
| Wife | 20.4 (20.3, 20.5) | 2.2 |
| Child | 20.7 (20.5, 20.9) | 2.2 |
| Other (Maid and relatives) | 19.8 (16.6, 23.1) | 2.6 |
| **Marital status** | | |
| Married | 20.4 (20.3, 20.5) | 2.2 |
| Single | 20.7 (20.5, 20.9) | 2.2 |
| others (Divorced and Widowed) | 20.2 (19.8, 20.7) | 2.3 |
| **Educational status** | | |
| Couldn't read and write | 20.2 (20.1, 20.3) | 2.3 |
| Read and write | 20.4 (20.1, 20.7) | 2.0 |
| Primary | 20.7 (20.6, 20.9) | 2.2 |
| Secondary and above | 20.7 (20.4, 20.9) | 2.2 |
| **Occupational status** | | |
| Farmer | 20.3 (20.2, 20.4) | 2.2 |
| Student | 20.8 (20.7, 21.0) | 2.2 |
| Trader | 21.0 (20.7, 21.3) | 2.6 |
| Other (employed, dependent, daily labourer) | 19.9 (19.5, 20.3) | 2.2 |
| **Membership of safety net program** | | |
| Yes | 20.3 (20.1, 20.5) | 2.0 |
| No | 20.5 (20.4, 20.5) | 2.3 |

model by considering its importance. On the other hand, all individual-level factors had a P-value of < 0.25. However, the final model did not include occupational status because its inclusion in the multivariable analysis increased the Akaike Information Criterion (AIC) from 5217 to 5222. Therefore, the determinant factors of women's BMI identified in the final model were household food insecurity score, household food consumption score, previous season BMI, age, educational status, marital status, and productive safety net program membership.

The analysis showed that as age increased by one year, women's BMI increased by β coefficient of 0.03 (95% CI: 0.02, 0.04). Household food insecurity score had an inverse relation with the BMI of women with β coefficient of -0.02 (95% CI: -0.03, -0.01), while the household food consumption score had a positive relation of 0.01 (95% CI: 0.01–0.02). Furthermore, the study found that women with a lower BMI in the previous season were likelier to have a lower BMI in the current season, as indicated by a β coefficient of 0.91 (95% CI: 0.89, 0.93). Unmarried women had higher BMI than married women, with a β coefficient of 0.57 (95% CI: 0.36, 0.77). Additionally, women who were not beneficiaries of the safety net program had higher BMIs than program beneficiaries, who had 0.38 (95% CI: 0.18–0.57); see Table 4.

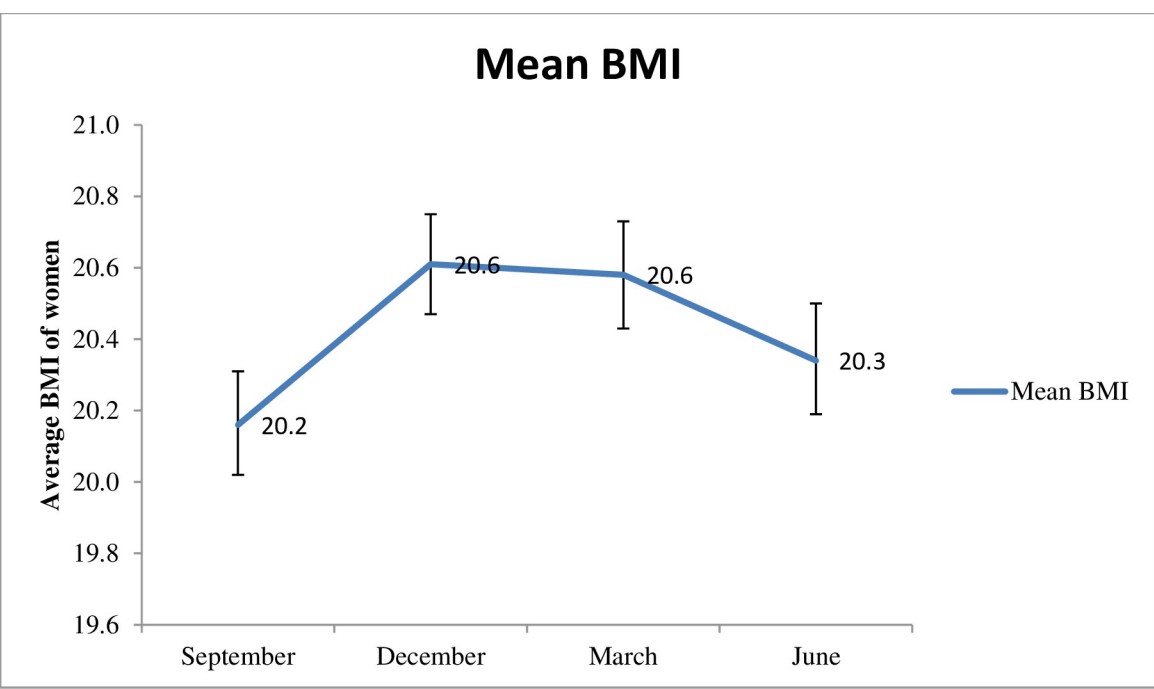

**Fig 3. Comparison of women's BMI along different seasons, from September 2021 to June 2022, South Ethiopia.** Seasonal variation in Women's Body Mass Index. This figure illustrates the seasonal variation in the average body mass index of women in the study population. The abbreviations used is BMI: Body Mass Index.

## Discussion

This study revealed the seasonal variation in household food access. Contrary to traditional assumptions, the study found that food scarcity occurred during the lean period in June, a month traditionally considered part of the food-plenty season in September. This suggests that

**Table 4. Predictors of BMI of women of reproductive age group from September 2021 to June 2022, South Ethiopia.**

| Variables | β coefficient | 95% CI | P-value |
|---|---|---|---|
| Age | 0.03 | 0.02–0.04 | <0.001* |
| Educational status | -0.01 | -0.03, 0.02 | 0.578 |
| Household food insecurity score | -0.02 | -0.03–0.01 | 0.004* |
| Household dietary diversity score | 0.03 | -0.02–0.09 | 0.224 |
| Food consumption score | 0.01 | 0.01–0.02 | <0.001* |
| Wealth index | 0.00002 | -0.0001–0.0001 | 0.595 |
| Previous BMI | 0.91 | 0.89–0.93 | <0.001* |
| **Marital status** | | | |
| Married | 1 | | |
| Single | 0.57 | 0.36–0.77 | <0.001* |
| others (Divorced and Widowed) | -0.14 | 0.47–0.2 | 0.416 |
| **Safety net** | | | |
| Yes | 1 | | |
| No | 0.38 | 0.18–0.57 | <0.001* |

* P- Value < 0.05

the timing and duration of food shortages are unpredictable, underscored by the need to understand food accessibility dynamics.

The study also showed a relative improvement in women's average BMI during the season of better food access, indicating their higher vulnerability to seasonal undernutrition. Lower BMI was observed among married women, Productive Safety Net Program (PSNP) beneficiaries, and those with lower BMI in the previous season, suggesting these groups face compounded challenges. However, in this study, seasonal change in household wealth was not a predictor of women's BMI.

These findings suggest the need to consider the potential influence of localized weather patterns and the importance of examining seasonal food shortages within the broader climatic context. The development of climate resilience initiatives might help ensure year-round household food accessibility. Additionally, comprehensive interventions to better serve the specific needs of the most susceptible women to seasonal undernutrition are required to make a tangible difference in safeguarding their health and well-being.

The study's methodological approach was a key strength; an open cohort design allowed us to capture the dynamic, time-dependent relationships between food access, household wealth, and women's nutritional status. This longitudinal perspective allowed the detection of trends and changes over time, a significant advancement from previous cross-sectional studies that provided only a static snapshot. The open cohort design also helped minimize loss to follow-up, enhancing the validity of the findings.

The main limitations of the study include the length of follow-up time, different kinds of biases, and residual confounders. In drought-prone areas like Boricha, the study period of one year may have limited the ability to map seasonal patterns fully. Secondly, there is a limitation regarding the BMI of adolescents in the last two seasons. Since height measurements were only taken during the first two consecutive seasons, the BMI of adolescents in later seasons may be higher than expected due to potential height changes within six months. This study might be susceptible to information bias, such as social desirability and recall bias. Moreover, other unmeasured factors could affect the findings, such as the impact of school feeding programs, deworming, parity, contraception use, and other variables that should have been accounted for in our study. Lastly, using an open cohort design restricts the establishment of causal relationships.

Despite the potential biases mentioned earlier, various efforts were undertaken to mitigate their impact on the study's validity. We tried to minimize the social desirability bias aroused by the potential over-reporting of food scarcity in anticipation of aid by clearly explaining the study's objectives to the respondents. Given that most dietary assessments relied on memory, there was a susceptibility to recall bias. To address this, we employed various strategies: we prompted respondents to recall their situations for household food security assessments and provided illustrative examples for each condition. Respondents were requested to list their food intake over the past 24 hours to minimize recall bias in household dietary diversity assessments. Furthermore, we managed recall bias in food consumption by restricting the duration of the food frequency analysis to a week. To mitigate information bias arising from inter- and intra-observer variability in anthropometric measurements, rigorous training sessions were conducted repeatedly. Despite the study's limitations, the sample size was sufficient for generalizing the findings to similar drought-prone areas.

In the current study, food availability was better in post-harvest than pre-harvest season, which aligns with findings in previous studies [9, 13, 59, 60]. A study conducted in the same area about five years ago showed that the prevalence of household food insecurity was higher in March than in September [34], contrary to the current study. The unexpected food shortage in September and better household food security in March might be related to the inadequate

and late onset of *Belg* rain for 2021 [44]. This might have postponed the harvest time, prolonged the lean period until September, and extended the food stock for longer. Alternatively, there could be unmeasured nutritional interventions, support, or aids that reduce the magnitude of household food insecurity in March [7]. The food accessibility in December and June followed the expected pattern.

The mean BMI is consistent with a study conducted in eastern Ethiopia [61] but slightly lower than other studies in Ethiopia [62–64]. The difference in average BMI in the later studies could be due to differences in the study target. The target groups in these studies were lactating women. The current study was also conducted in drought-prone areas, which could contribute to the lower average BMI. This relatively lower average BMI suggests the presence of potential nutritional deficiencies and associated health risks.

In our study, women's average BMI was higher during December and March, corresponding to seasons with relatively better food access. This observation aligns with previous studies demonstrating the positive impact of improved household food access on women's nutritional status [14, 35, 65].

Food-insecure households may not get adequate amounts and quality of food, predisposing them to undernutrition or overnutrition [66]. In this study, household food security and high food consumption scores were associated with better BMI, consistent with other study findings [36, 47, 67]. This study's mean BMI improved as age increased, which is consistent with previous studies [68, 69]. As women age increases, reproductive responsibilities also increase, possibly contributing to weight gain.

In the current study, unmarried women had a better BMI than married women, supported by a few studies [70, 71]. Most unmarried women were 15 to 19 years old and were primary school students, potentially receiving additional food from school feeding programs [72]. On the other hand, almost all married women were mothers, and food allocation was their responsibility. Therefore, during times of food shortage, they might prioritize their children [73]. This could be an additional explanation for the higher BMI among unmarried women in the current study. This suggests that mothers are more susceptible to seasonal undernutrition compared to women who are not mothers.

The Productive Safety Net Program (PSNP) is one of the social protection schemes in Ethiopia [74]. It aims to support vulnerable households by offering cash transfers or food assistance for half of the year, usually during food shortages. However, several challenges hinder the program's effectiveness, including inadequate cash transfers, delayed payments, and lack of sustainable effect [75]. Despite these challenges, the PSNP positively impacts household food security and helps to prevent asset loss during catastrophic shocks [48, 75, 76]. The current study indicated that the mean BMI of PSNP beneficiary women was lower than that of the non-beneficiaries, as shown by [61].

Contrary to this, other studies have shown that PSNP reduces childhood stunting and women's undernutrition [64, 77]. In the current study, the BMI of PSNP beneficiaries was lower than their counterparts in all seasons. This could suggest a pre-existing difference in average BMI between PSNP beneficiaries and non-beneficiaries. Therefore, further research is required to determine the effect of the PSNP on women's nutritional status. This finding indicates that the Safety Net program should reassess the adequacy of its support to its beneficiaries. Furthermore, it should be integrated with other interventions to enhance the program's effectiveness.

This study found that women's nutritional status in the previous season influences their nutritional status in the current season. This suggests that undernourished women will likely remain undernourished, regardless of some improvement in food availability.

## Conclusion

Household food security, dietary diversity, and food consumption vary across seasons. These variations are accompanied by changes in women's BMI, highlighting their vulnerability to undernutrition during certain seasons. However, the patterns of variation are different from year to year and are influenced by the weather conditions. Any deviations from typical weather patterns can decrease crop productivity and food accessibility and subsequently impact the nutritional status of the women. In addition to food accessibility, age, marital status, productive safety net membership, and previous BMI were predictors of women's BMI.

## Recommendation

To tackle seasonal food shortages and their disproportionate impact on women's nutrition, we recommend that policymakers integrate sustainable food access policies across sectors and develop gender-responsive social protection programs. Local governments should empower women through entrepreneurship, vocational training, financial services, and nutrition education. Researchers should explore options to minimize reliance on rain-fed agriculture and evaluate the effectiveness of social protection programs. Farmers should diversify their income sources by engaging in off-farm activities. Implementing these recommendations could improve the nutritional well-being of women and other vulnerable groups.

## Supporting information

**S1 Table. Socio-demographic characteristics of 15 to 49-year-old women in South Ethiopia, June 2021.** This table provides the socio-demographic information of 1,089 women participants from 894 households. The data are presented as count (n) and percentage (%). The abbreviation "N" refers to the total number of participants.
(DOCX)

**S2 Table. Socio-demographic characteristics of household heads, South Ethiopia, June 2021.** This table presents the socio-demographic information of the household heads where the study participants reside. The data are presented as count (n) and percentage (%).
(DOCX)

## Acknowledgments

We express our gratitude to the COGENT project for funding this study. We want to thank the Health Bureau of the Sidama National Regional State and the health offices of the Boricha and Bilate Zuria districts for their valuable support. We would also like to thank the study participants and data collectors for their contributions. Lastly, we acknowledge the data clerk for their assistance in data entry.

## Author Contributions

**Conceptualization:** Bethelhem Mezgebe, Taye Gari, Mehretu Belayneh, Bernt Lindtjørn.

**Data curation:** Bethelhem Mezgebe, Taye Gari, Mehretu Belayneh, Bernt Lindtjørn.

**Formal analysis:** Bethelhem Mezgebe, Taye Gari, Mehretu Belayneh, Bernt Lindtjørn.

**Funding acquisition:** Bernt Lindtjørn.

**Investigation:** Bethelhem Mezgebe, Taye Gari, Mehretu Belayneh, Bernt Lindtjørn.

**Methodology:** Bethelhem Mezgebe, Taye Gari, Mehretu Belayneh, Bernt Lindtjørn.

**Project administration:** Taye Gari.

**Resources:** Taye Gari, Bernt Lindtjørn.

**Software:** Bethelhem Mezgebe, Taye Gari, Mehretu Belayneh, Bernt Lindtjørn.

**Supervision:** Taye Gari, Mehretu Belayneh, Bernt Lindtjørn.

**Visualization:** Bethelhem Mezgebe, Taye Gari, Mehretu Belayneh, Bernt Lindtjørn.

**Writing – original draft:** Bethelhem Mezgebe, Taye Gari, Mehretu Belayneh, Bernt Lindtjørn.

**Writing – review & editing:** Bethelhem Mezgebe, Taye Gari, Mehretu Belayneh, Bernt Lindtjørn.

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
