## [Decision Letter · Decision Letter 0]

1 Apr 2024

PGPH-D-24-00274

Seasonal variations in household food insecurity and consumption affect women’s nutritional status in rural South Ethiopia.

Dear Dr. Mezgebe,

Thank you for submitting your manuscript to PLOS Global Public Health. After careful consideration, we feel that it has merit but does not fully meet PLOS Global Public Health’s publication criteria as it currently stands. Therefore, we invite you to submit a revised version of the manuscript that addresses the points raised during the review process.

EDITOR: Please insert comments here and delete this placeholder text when finished. Be sure to:

Please ensure that your decision is justified on PLOS Global Public Health’s publication criteria and not, for example, on novelty or perceived impact.==============================Please submit your revised manuscript by 22/04/2024. If you will need more time than this to complete your revisions, please reply to this message or contact the journal office at globalpubhealth@plos.org. Please include the following items when submitting your revised manuscript:
A rebuttal letter that responds to each point raised by the editor and reviewer(s). You should upload this letter as a separate file labeled 'Response to Reviewers'.A marked-up copy of your manuscript that highlights changes made to the original version. You should upload this as a separate file labeled 'Revised Manuscript with Track Changes'.An unmarked version of your revised paper without tracked changes. You should upload this as a separate file labeled 'Manuscript'.

We look forward to receiving your revised manuscript.

Kind regards,

Dickson Abanimi Amugsi, PhD

Academic Editor

Journal Requirements:

If you did not receive any funding for this study, please simply state: “The authors received no specific funding for this work.

Additional Editor Comments (if provided):

Reviewers' comments:

Reviewer's Responses to Questions

**Comments to the Author**

1. Does this manuscript meet PLOS Global Public Health’s publication criteria? Is the manuscript technically sound, and do the data support the conclusions? The manuscript must describe methodologically and ethically rigorous research with conclusions that are appropriately drawn based on the data presented.

Reviewer #1: Yes

Reviewer #2: Yes

Reviewer #3: Yes

2. Has the statistical analysis been performed appropriately and rigorously?

Reviewer #1: Yes

Reviewer #2: No

Reviewer #3: Yes

3. Have the authors made all data underlying the findings in their manuscript fully available (please refer to the Data Availability Statement at the start of the manuscript PDF file)?

Reviewer #1: Yes

Reviewer #2: Yes

Reviewer #3: Yes

4. Is the manuscript presented in an intelligible fashion and written in standard English?

Reviewer #1: Yes

Reviewer #2: Yes

Reviewer #3: No

5. Review Comments to the Author

Reviewer #1: Thank you for preparing the article. I have a few comments which I hope would help with improving the manuscript.

Title

I suggest the authors consider rewriting the objective of the study for clarity. “Food insecurity,” which has a negative connotation, is not consistent with “dietary diversity” and “food consumption” and makes the objective hard to read. The objective can be refined to: “To assess the effect of seasonal variation in household food security, adequate dietary diversity and food consumption, and household income on the nutritional status of women of reproductive age in a rural community in South Ethiopia. Further, the study aimed at identifying associated factors to women's nutritional status”.

Introduction

It would be helpful to the reader if the authors specified in the introduction what information is from Ethiopia and what is from other countries.

Study setting and context

What was the reason for using “live birth rate” instead of “birth rate”? “Live birth rate is usually used in fertility reporting and not at the population level.

Selection of study population and sample size

Could you add the total number of women included at the end of the section and specify the reason for including all eligible women?

Wealth index

Line 217: typo. “Cooking”

Meal frequency

I couldn’t find the description of calculating “meal frequency” in the methods. Further, if data is available, including the meal frequency finding in Table 2 would be informative.

Discussion

I suggest the authors discuss the significance of the findings; the discussion primarily compares the findings of the study with other studies. However, the validity and significance of the findings could have been discussed further. I also would like authors to discuss the implications of funding, not only for future research but also in designing development projects.

Reviewer #2: Dear author,

Thank you for interesting in this area, which needs great attention all sectors to overcome the problem of malnutrition. However, a lots of researches have been conducted and published on this title. So, the followings are the main questions raised on your work. I have included the comments also.

1. what a new thing your research findings contribute to the science?

2. the recommendation needs to be revised and inline with the study findings.

3. Use the journal guidelines as all contents of the paper has poor write up.

4. Use the scientific tables for the study results

5. Your discussion is weak and recommendations also need revision.

Reviewer #3: The manuscript entitled "Seasonal variation in household food insecurity and consumption affect women's nutritional status in rural South Ethiopia" is important as it deals with with household food security and its impact to women's health. The study findings will be useful to Policymakers, farmers, health officials, extension officers, government, researchers, students, other readers, etc. I suggest the following to improve the quality of the manuscript.

1. The Author must include the hypothesis towards the end of Introduction section.

2. The Methods and Materials section is not organised properly and needs to be rearranged. The first subsection must be "Study area description, food production and food availability". The current subsections "Study design" in Line 107 and Study setting and context in Line 110 must be deleted. This means that the content under these deleted subsection will be under new subsection.

3. The subsection "Ethical consideration must be second subsection following Study area description, food production and availability

4. The subsection "Variables" in Line 159 must be deleted and be replaced with subsection "Data collection procedures". All the content before Data processing and analysis will be under new subsection of "Data collection procedures"

5. It will be interesting for the Author to make recommendation(s) to Policymakers based on this study findings to ensure sustainable nutritious food availability throughout the year including food scarcity seasons.

6. There are some minor gramatical errors and the Author needs to go through the manuscript, reading it carefully and correct these.

6. PLOS authors have the option to publish the peer review history of their article (what does this mean?). If published, this will include your full peer review and any attached files.

**Do you want your identity to be public for this peer review?** For information about this choice, including consent withdrawal, please see our Privacy Policy.

Reviewer #1: **Yes: **Behnoush Ahranjani

Reviewer #2: **Yes: **Tamiru Yazew

Reviewer #3: No

---

## [Decision Letter · Decision Letter 1]

5 Jun 2024

PGPH-D-24-00274R1

Seasonal variations in household food security and consumption affect women’s nutritional status in rural South Ethiopia.

Dear Bethelhem,

Thank you for submitting your manuscript to PLOS Global Public Health. After careful consideration, we feel that it has merit but does not fully meet PLOS Global Public Health’s publication criteria as it currently stands. Therefore, we invite you to submit a revised version of the manuscript that addresses the points raised during the review process.

We look forward to receiving your revised manuscript.

Kind regards,

Dickson Abanimi Amugsi, PhD

Academic Editor

Journal Requirements:

Additional Editor Comments (if provided):

Reviewers' comments:

Reviewer's Responses to Questions

**Comments to the Author**

1. If the authors have adequately addressed your comments raised in a previous round of review and you feel that this manuscript is now acceptable for publication, you may indicate that here to bypass the “Comments to the Author” section, enter your conflict of interest statement in the “Confidential to Editor” section, and submit your "Accept" recommendation.

Reviewer #1: (No Response)

Reviewer #3: All comments have been addressed

2. Does this manuscript meet PLOS Global Public Health’s publication criteria? Is the manuscript technically sound, and do the data support the conclusions? The manuscript must describe methodologically and ethically rigorous research with conclusions that are appropriately drawn based on the data presented.

Reviewer #1: Partly

Reviewer #3: Yes

3. Has the statistical analysis been performed appropriately and rigorously?

Reviewer #1: I don't know

Reviewer #3: Yes

4. Have the authors made all data underlying the findings in their manuscript fully available (please refer to the Data Availability Statement at the start of the manuscript PDF file)?

Reviewer #1: Yes

Reviewer #3: Yes

5. Is the manuscript presented in an intelligible fashion and written in standard English?

Reviewer #1: No

Reviewer #3: Yes

6. Review Comments to the Author

Reviewer #1: Thank you for addressing the comments and updating the manuscript. Based on the updated manuscript, I have some recommendations.

The updated introduction section is long and can be rewritten to be more concise. Further, some detailed information about other studies can be used in writing the discussion and not being included in the introduction.

Could you please check that "paid activities" L88 refers to paid and not unpaid?

L285, I believe taking height measures in adults for two times addresses reliability or accuracy of measures and not validity.

Did you use dietary recall to assess food consumption? If so and you haven't mentioned it somewhere, could you add it?

Table 3: numbers don't add up. Please check and if this is because of missing data, note that.

Discussion:

-Generally, before comparing the findings of the study to other studies, the importance and significance of the findings should be discussed. How accurate and relevant are the findings? I recommend re-writing the discussion.

- What does "weighted" food consumption mean?

- Regarding PSNP, when was the BMI of beneficiaries were lower than their peers? I wonder if the recipients of PSNP had lower BMI than others and that's the reason they were part of the program.

- Please check L481: is there any evidence to suggest the impact of changes in seasonal wealth on women's nutrition status?

-L503: What does information bias include?

Recommendation:

What is the difference between year-round access and sustainable access? I recommend merge these two categories and group the recommendations under different stakeholders such as policy makers, local government, etc.

I recommend a final edit of the document for English language to prepare a concise version.

Reviewer #3: (No Response)

7. PLOS authors have the option to publish the peer review history of their article (what does this mean?). If published, this will include your full peer review and any attached files.

**Do you want your identity to be public for this peer review?** For information about this choice, including consent withdrawal, please see our Privacy Policy.

Reviewer #1: **Yes: **Behnoush Ahranjani

Reviewer #3: No

---

## [Decision Letter · Decision Letter 2]

16 Jul 2024

Seasonal variations in household food security and consumption affect women’s nutritional status in rural South Ethiopia.

PGPH-D-24-00274R2

Dear Dr Mezgebe,

We are pleased to inform you that your manuscript 'Seasonal variations in household food security and consumption affect women’s nutritional status in rural South Ethiopia.' has been provisionally accepted for publication in PLOS Global Public Health.

Best regards,

Dickson Abanimi Amugsi, PhD

Academic Editor

Please work with the editorial office to address this very minor issues from reviewer 2:

Thank you for addressing the comments.

Please check L91: it seems it should be association and not impact, and L455: incomplete sentence.

The manuscript needs editing for English.

Thank you

Reviewer Comments (if any, and for reference):

Reviewer's Responses to Questions

**Comments to the Author**

1. If the authors have adequately addressed your comments raised in a previous round of review and you feel that this manuscript is now acceptable for publication, you may indicate that here to bypass the “Comments to the Author” section, enter your conflict of interest statement in the “Confidential to Editor” section, and submit your "Accept" recommendation.

Reviewer #1: All comments have been addressed

2. Does this manuscript meet PLOS Global Public Health’s publication criteria? Is the manuscript technically sound, and do the data support the conclusions? The manuscript must describe methodologically and ethically rigorous research with conclusions that are appropriately drawn based on the data presented.

Reviewer #1: Partly

3. Has the statistical analysis been performed appropriately and rigorously?

Reviewer #1: I don't know

4. Have the authors made all data underlying the findings in their manuscript fully available (please refer to the Data Availability Statement at the start of the manuscript PDF file)?

Reviewer #1: (No Response)

5. Is the manuscript presented in an intelligible fashion and written in standard English?

Reviewer #1: No

6. Review Comments to the Author

Reviewer #1: Thank you for addressing the comments.

Please check L91: it seems it should be association and not impact, and L455: incomplete sentence.

The manuscript needs editing for English.

7. PLOS authors have the option to publish the peer review history of their article (what does this mean?). If published, this will include your full peer review and any attached files.

**Do you want your identity to be public for this peer review?** For information about this choice, including consent withdrawal, please see our Privacy Policy.

Reviewer #1: **Yes: **Behnoush Ahranjani
